# Defining Community-Acquired Pneumonia as a Public Health Threat: Arguments in Favor from Spanish Investigators

**DOI:** 10.3390/medsci8010006

**Published:** 2020-01-25

**Authors:** Catia Cillóniz, Rosario Menéndez, Carolina García-Vidal, Juan Manuel Péricas, Antoni Torres

**Affiliations:** 1Department of Pneumology, Hospital Clinic of Barcelona, Center for Biomedical Research Networking Centers in Respiratory Diseases (Ciberes), August Pi i Sunyer Biomedical Research Institute (IDIBAPS), University of Barcelona, 08036 Barcelona, Spain; catiacilloniz@yahoo.com; 2Department of Pneumology, Hospital Universitario y Politécnico La Fe/Instituto de Investigación Sanitaria (IIS) La Fe, Center for Biomedical Research Network in Respiratory Diseases (Ciberes), 46026 Valencia, Spain; rosmenend@gmail.com; 3Infectious Diseases Department, Hospital Clinic of Barcelona, 08036 Barcelona, Spain; carolgv75@hotmail.com; 4Clinical Direction of Infectious Diseases and Clinical Microbiology of Lleida, Hospital Universitari Arnau de Vilanova, Translational Research Group on Infectious Diseases of Lleida (TRIDLE), IRB Lleida, 25198 Lleida, Spain; jmpericas.lleida.ics@gencat.cat

**Keywords:** pneumonia, community-acquired pneumonia, pneumonia burden, pneumonia epidemiology

## Abstract

Despite advances in its prevention, pneumonia remains associated with high morbidity, mortality, and health costs worldwide. Studies carried out in the last decade have indicated that more patients with community-acquired pneumonia (CAP) now require hospitalization. In addition, pneumonia management poses many challenges, especially due to the increase in the number of elderly patients with multiple comorbidities, antibiotic-resistant pathogens, and the difficulty of rapid diagnosis. In this new call to action, we present a wide-ranging review of the information currently available on CAP and offer some reflections on ways to raise awareness of this disease among the general public. We discuss the burden of CAP and the importance of attaining better, faster microbiological diagnosis and initiating appropriate treatment. We also suggest that closer cooperation between health professionals and the population at large could improve the management of this largely preventable infectious disease that takes many lives each year.

## 1. Introduction

*Pneumonia* can *affect* people *of all ages* and is a major public health problem worldwide. It is a significant cause of mortality and morbidity, especially in children under five years of age and in older adults. The Global Burden of Diseases (GBD), Injuries, and Risk Factors Study 2016 [1] reported that lower respiratory tract infections (LRTI) affected over 336 million people all over the world, causing an estimated 65.9 million hospitalizations and 2,377,697 deaths. Interestingly, the GBD report in 2017 [2] showed that the total deaths from LRTI had fallen by 36% between 2007 and 2017 in children under the age of five but had risen by 34% among adults aged 70 or over. *Streptococcus pneumoniae* continues to be the leading global cause of LRTI morbidity and mortality, contributing to more deaths than all other etiologies combined in 2016 [1]. Although mortality due to pneumococcal pneumonia in the under-fives fell between 1990 and 2017, it increased among older adults [2]. Recently, there has been a growing interest in the interaction of bacteria and respiratory viruses in the pathogenesis of pneumonia. Respiratory viruses may be the primary cause of severe pneumonia, but the condition may also present in conjunction with (or be followed by) a secondary bacterial infection, most commonly caused by *S. pneumoniae*, *Staphylococcus aureus*, or *Pseudomonas aeruginosa*, which may colonize the host’s respiratory tract. Infection with the influenza virus is associated with a higher risk of bacterial sepsis and acute respiratory distress syndrome [3].

Data from the GBD in Spain showed that pneumonia was the 16th most important cause of mortality in 2016, leading to 22,308 deaths [4]. Unfortunately, pneumonia is not a reportable disease, and therefore its true incidence is unknown. However, using data from prospective observational studies, we can put together a broad picture of the impact of community-acquired pneumonia (CAP) in Spain. The overall incidence of CAP in adults in the Spanish primary care setting between 2009 and 2013 [5] was estimated to be 4.63 cases per 1000 persons/year, increasing by age group (from 1.98 in the 18–20 year age group to 23.74 in the ≥90 year group), and was higher in males (5.04) than in females (4.26). The risk factors related to CAP in primary care were HIV infection (OR 5.21; 95% CI: 4.35 to 6.27), chronic obstructive pulmonary disease (COPD) (OR 2.97; 95% CI: 2.84 to 3.12), asthma (OR 2.16; 95% CI: 2.07 to 2.26), smoking (OR 1.96; 95% CI: 1.91 to 2.92), and poor dental hygiene (OR 1.45; 95% CI: 1.41 to 1.49) [6].

Other studies showed that the incidence of CAP in Spanish adults requiring hospitalization rose from 1.42 cases per 1000 persons/year in 2004 to 1.63 cases in 2013. The incidence increased in both sexes (from 2.08 to 2.27 in males and from 0.92 to 1.16 in females) and in the older age groups (from 10.06 to 11.00 in the 75–84 year group, and from 2.11 to 25.84 in the ≥85 year group) [7]. Interestingly, two recent studies showed an increase in the incidence of CAP in COPD patients (from 13.44 per 1000 persons/year in 2004–2005 to 16.40 cases in 2012–2013) and in patients with diabetes (from 8.14 per 1000 persons/year in 2004 to 9.23 cases in 2013); COPD and diabetes are both important and frequent comorbidities in CAP patients [7,8,9].

Pneumonia has a particularly strong impact on vulnerable populations. CAP is frequent among refugees [10], the homeless [11,12,13,14], drug users [15,16] and people of low socioeconomic status [17] and is associated with increased mortality.

The rapid emergence of antimicrobial resistance among pneumonia-causing bacteria all over the world has forced the scientific community to reflect on the advantages and disadvantages of antibiotic treatment of CAP [18] (Figure 1). Approximately 6% of cases of CAP with etiological diagnosis are caused by multidrug-resistant (MDR) pathogens, the most frequently described being *S. aureus* and *P. aeruginosa* [19,20,21]. A recent multinational point prevalence study reported that the prevalence rates of drug-resistant *S. pneumoniae* and MDR Enterobacteriaceae in CAP were 1.3% and 1.2%, respectively [22,23]. MDR pathogens make the clinical management of CAP a real challenge for physicians.

The causes of the growing use of antibiotics and the non-adequacy of targeted therapies against microorganisms are diverse and complex. Many clinical studies have demonstrated the efficacy of these antibiotics in terms of reducing mortality and complications (also when used in combined therapy), but their ecological impact has been largely neglected. Moreover, our diagnostic approach to CAP is limited; improvements are urgently needed, such as the introduction of new molecular assays for rapid microbiological identification [24]. In this connection, the study by Gadsby et al. [25] used a quantitative multipathogen (bacteria and virus) molecular assay to determine the etiology of CAP in hospitalized patients and was able to identify the cause of pneumonia in 87% of cases, compared with a rate of only 39% when using microbiological culture alone. The authors also concluded that the PCR results would have been a useful tool for antibiotic de-escalation in 77% of the cases of CAP. The results of that study demonstrate the value of these new methods for the diagnosis of pneumonia.

In this new call to action, we present a wide-ranging review of the information currently available on CAP and offer some reflections on ways to raise awareness of this disease among the general public. We discuss the burden of CAP and the importance of attaining better and faster microbiological diagnosis and initiating appropriate treatment. We also suggest that closer cooperation between health professionals and the population at large could improve the management of this largely preventable infectious disease that takes many lives each year.

## 2. CAP Affects Everyone: Why Do We Not Recognize the Threat?

In Spain, there is a general lack of knowledge about CAP and, particularly, of its impact on population’s health. There is a widespread misperception that CAP affects only specific subgroups of susceptible individuals: in some people, CAP may be associated with only a few clinical symptoms but in others, it may present as a fulminant systemic infection causing respiratory failure, multiple organ dysfunction, and death [26,27]. In a German study assessing knowledge of sepsis among the elderly population (≥65 years old) through a telephone survey of 1401 persons, 89% had heard the term “sepsis” before the survey, but 39% could not identify a correct definition, while 98% were familiar with the term “blood poisoning” [28]. Interestingly, 45% responded that sepsis is an intense allergic reaction, and 30% that it can be caused by MDR pathogens in hospitals. Roughly 18% of respondents knew that infections such as influenza or CAP can cause sepsis. Although 83% of interviewees were aware that sepsis is a severe condition requiring urgent medical assistance, they underestimated its mortality, with 50% responding that mortality due to acute myocardial infarction is higher than that due to sepsis. As for vaccination, only 17% of the participants were aware that it reduces the risk of sepsis [9]. This latter finding corroborates those of previous studies which reported that the perception of the severity of pneumonia was associated with receiving vaccination, especially amongst elderly persons [29,30,31].

Montull et al. reported that one-third of CAP cases in Spain present with severe sepsis [32]. In that study, elderly patients, alcohol abusers, patients with renal disease, and COPD patients were found to be more likely to develop sepsis, and *S. pneumoniae* and mixed etiology were the main causes of severe sepsis. A recent Spanish study [33] investigating the risk and prognostic factors in very old (≥80 years) CAP patients who developed sepsis reported a prevalence of 71% using Sepsis-3 criteria and identified male sex, diabetes mellitus, and chronic renal disease as associated risk factors for sepsis. The mortality rates at 30 days and 1 year were 15% and 30%, respectively.

The misconception of the severity and complications of CAP among the general public is alarming [34]. CAP patients have a four-fold increased risk of cardiac events [35] such as acute myocardial infarction, cardiac arrhythmia, and new-onset heart failure, both within 30 days of admission and up to 10 years after the CAP episode [35,36,37]. The risk factors associated with cardiac events in CAP are infection with *S. pneumoniae*, older age, severe pneumonia, obesity, hyperlipidemia, and arterial hypertension [35,38,39]. Between 10% and 30% of patients with bacteremic pneumococcal CAP present cardiovascular events, the most affected being those with preexisting cardiovascular disease [38,39,40]. A possible explanation for this cardiac tissue damage is that the substances produced, including Damage-Associated Molecular Patterns (DAMPs), may maintain inflammation during severe lung tissue cell injury, thus affecting other host cells and aggravating heart cell injury [41].

Importantly, survivors of CAP have a substantially increased likelihood of dying long after recovery from the acute episode (i.e., more than 10 years later), even patients with no previous comorbidities [42,43,44]. Unfortunately, the general public’s lack of awareness of the severity of pneumonia and its consequences in the short and long term is mirrored by the unacceptably low use of influenza and pneumococcal vaccines in our country [45,46]; the discrepancies between the recommendations for pneumococcal vaccination in the document recently published by the Spanish Ministry of Health [47] and the consensus document on pneumococcal vaccination in adults signed by 18 scientific societies [48] only make the problem worse.

## 3. The Role of Pneumonia and Secondary Bacterial Infection in the H1N1 Influenza A Pandemic

A decade has passed since the 2009 H1N1 influenza pandemic which caused approximately 201,200 deaths due to respiratory diseases, principally pneumonia, and an additional 83,300 deaths due to cardiovascular diseases in the first year of the circulation of the virus. Significantly, most deaths occurred in young adults who often presented no comorbidities [49,50].

A largely unknown feature of this pandemic is the fact that pneumonia was identified in one out of four patients, with influenza virus H1N1, *S. pneumoniae*, and *S. aureus* being the most frequently detected microorganisms [51,52,53,54,55]. Although most H1N1 influenza cases presented with mild symptoms, approximately 25% developed severe acute respiratory distress syndrome (ARDS) requiring intensive care unit (ICU) admission, and in many cases these patients were treated with extracorporeal membrane oxygenation (ECMO) [56,57].

The association of *Aspergillus* spp. pulmonary infections and influenza infection was another notable finding, especially in critically ill patients [58]. Regrettably, the mortality rate in these cases was approximately 40%; immunocompromised patients were mainly affected, but mortality was also reported in apparently immunocompetent patients treated with corticosteroids [59].

The lessons we learned from the 2009 H1N1 influenza pandemic included the importance of hospital preparedness, the key role of the organization of clinical care in multidisciplinary teams, the importance of early diagnosis and treatment of influenza A H1N1 cases, and, above all, the role of influenza and pneumococcal vaccination. Interestingly, recent studies report that antiviral agents for influenza infection, such as oseltamivir, have a limited effect on the prevention of the onset or progression of pneumonia [60]. Through the study of the epidemiological and clinical characteristics of the 2009 H1N1 pandemic, we have learned that the pathogenesis of influenza is not a virus-induced cytopathy but a hyperimmune reaction of the host (such as a cytokine storm) against the influenza virus infection. In addition, some investigators have suggested that timely early immune-modulator therapy was able to prevent disease progression in the 2009 influenza pandemic.

Since 2009, the distinctive features of polymicrobial CAP such as inflammatory and host response as well as disease-related characteristics have also been emphasized. In a cohort of 362 CAP patients admitted to the ICU, 11% presented polymicrobial CAP, and the presence of chronic respiratory disease and ARDS were identified as risk factors for polymicrobial pneumonia. Polymicrobial pneumonia was a risk factor for inappropriate empiric antimicrobial therapy, which independently predicted hospital mortality [61].

There are many clinically relevant microbial interactions in pneumonia, for instance the interactions between the influenza virus and *S. pneumoniae*, between *S. pneumoniae* and *S. aureus,* and between the influenza virus and *S. aureus*. Initial CAP may be more likely to be caused by pathogens that do not respond to antibiotics, such as respiratory viruses, than by bacterial pathogens, and severe CAP caused by these pathogens may induce secondary bacterial invasion by agents such as *S. pneumoniae, S. aureus, and P. aeruginosa* that may colonize the host respiratory tract. We need to understand them better in order to improve CAP management [62,63,64,65,66].

The influenza A H1N1 pandemic is not the only phenomenon that has left us with lessons to learn. The tobacco epidemic also represents a huge problem. Tobacco smoking causes morphological changes in the epithelium of the bronchial mucosa, with the loss of cilia, mucous gland hypertrophy, and increased goblet cells [67]. Smoking is associated with the colonization of the lung by pathogenic bacteria and an increased risk of lung infections, especially in the case of colonization by *S. pneumoniae* [68,69]. The study by Bello et al. [70] concluded that current smokers with pneumococcal CAP often develop severe sepsis and require hospitalization at a younger age, despite having fewer comorbid conditions. In the CAP population, smoking increases the risk of 30-day mortality independently of tobacco-related comorbidity and age. A recently published systematic review reported that tobacco smoke exposure is significantly associated with the development of CAP in both current and ex-smokers and that passive smokers aged >65 years are at higher risk of CAP [71].

Environmental pollution is another important risk factor for respiratory infections like pneumonia. The recent study by Pirozzi et al. [72], which examined the relationships between short-term air pollution exposure and pneumonia severity, found particle levels greater than 12.5 μg/m^3^ to be associated with more cases of pneumonia, more hospital admissions, more severe cases of pneumonia, and higher hospital mortality.

## 4. The Development of New Technologies

To quote Chanderraj and Dickson [73], “Pneumonia remains a 21st century problem treated with 20th century therapies and diagnosed using 19th century tools”. It would be hard to find a better definition (Figure 2).

Several key recommendations in the latest international and national guidelines for the management of CAP are based on expert opinions [74,75,76]. One of them concerns the microbiological diagnosis of CAP. Arguably, the detection, identification, and determination of antimicrobial susceptibility patterns of microorganisms causing CAP are the main steps in establishing appropriate therapy, preventing the emergence of antimicrobial resistance by avoiding antibiotic selection pressure, and reducing health-associated costs deriving from CAP. In recent decades, diagnostic methods have made significant progress; this is especially true of molecular diagnostic techniques that reduce the response time to the etiologic diagnosis of pneumonia, help distinguish between bacterial and viral infection, and provide information on susceptibility to antimicrobials [24].

The value of these new technologies is reflected in Priest et al. study [77] of the performance of routine systematic PCR testing to calculate the incidence of Legionnaires’ disease in New Zealand hospitals. The overall incidence of Legionnaires’ disease reported in the study was 0.54/1000 persons/year; interestingly, *Legionella longbeachae* was the main cause (63%) of the cases not detected by the *Legionella* urinary antigen test. The results of that study show that the real burden of Legionnaires’ disease seems to have been underestimated and provide valuable information for future CAP management guidelines.

New technologies can be applied to different samples and patient populations, and the rapid availability of the results makes them an important diagnostic tool for clinicians managing CAP patients, especially the critically ill. However, these new technologies have not yet been extensively implemented in routine clinical practice, in part because of the difficulty in discriminating between colonization and infection in many microorganisms. The detection of microorganisms in the upper or lower respiratory tract is not a direct etiologic cause of lung cell injury in pneumonia and does not reflect a systemic immune reaction against the pathogens. We need more evidence from new approaches using quantitative methods to determine reliable cut-off levels. There is also an urgent need for more studies on the cost–effectiveness of these new technologies for diagnosing pneumonia, which might support their implementation in daily clinical practice [24].

## 5. New Concepts, When Times Change

Although weakened immunity is seen as an important factor in the pathogenesis of CAP, the potential clinical applications of immunological profiling for improving severity assessment and prognosis in CAP have not been extensively studied to date.

In 2017, a Spanish study [78] described an immunological profile in CAP patients called lymphopenic-CAP (L-CAP), which increases the risk of mortality in hospitalized CAP patients. These results were corroborated by subsequent studies that sought to explain this association [79,80]. One of these studies reported that the L-CAP immunophenotype might be useful for the assessment of pneumonia severity; the authors found that L-CAP is characterized by CD4^+^ depletion, has a higher inflammatory response, and presents lower IgG2 levels correlating with worse prognosis [80] (Table 1).

Lymphopenia may be a characteristic of severe pneumonia patients infected with respiratory pathogens (such as respiratory viruses, *mycoplasma pneumoniae*). Its severity is correlated with the severity of lung injury. The autopsy findings of severe ARDS patients and experimental animals infected with influenza viruses show lymphocyte depletion of whole lymphoid tissues. These data, together with lymphocyte predominance in early lung lesions, suggest that T cells may control the effects of substances from pathogens and/or injured host cells. Animals with depressed T cell function or loss of T cell function, such as nude mice, show milder or fewer pneumonia lesions than immune-competent animals in *Mycoplasma* or influenza virus infection models, although the time taken to detect pathogens in the lungs of animals with compromised T cells is longer. These findings suggest that lung injury is associated with T cell activation rather than with pathogen-induced cytopathy. Possibly, the capacity of the host immune system to mobilize immune cells against these relentless substances and to counter extensive lung cell injury in immune-competent patients may be restricted; patients with underlying disease or immune-deficient states may have a limited repertoire of immune cells. Furthermore, severe pneumonia or ARDS caused by a viral infection tends to induce subsequent bacterial infection from normal flora in patients, which increases the workload of the immune cells [81].

Although the immunopathogenesis of pneumonia (infectious or otherwise) remains unknown, it has been proposed that inflammation-inducing substances are involved in both forms of the disease. The host immune system may control these etiologic substances, not only those originating from pathogens such as toxins and pathogen-associated molecular patterns (PAMPs) but also those originating from injured or infected host cells, including damage-associated molecular patterns (DAMPs), pathogenic proteins, and pathogenic peptides. This is especially the case in intracellular pathogen infections such as virus, chlamydia, legionella, and possibly *Mycoplasma* infections (this is known as the protein-homeostasis-system hypothesis). When these substances spread systemically and locally and bind to target organ cells, clinical symptoms emerge due to the activation of corresponding immune cells and immune proteins. The substances produced from injured host cells induce further inflammation if released into the systemic circulation or near local lesions. The severity or chronicity of pneumonia or ARDS depends on the amount of etiologic substances with their corresponding immune reactions, the duration of the appearance of specific immune cells, or the repertoire of specific immune cells that control the substances. Therefore, the early control of lung injuries from initial hyperactive immune reactions performed by non-specific adaptive immune cells is crucial for reducing morbidity rates and preventing pneumonia progression in patients with CAP, including severe influenza pneumonia, *Mycoplasma* pneumonia, and other types. Thus, it has been proposed that earlier dose-adjusted corticosteroid therapy with an antibiotic can reduce the morbidity of pneumonia and prevent disease progression [82].

Future studies that simultaneously evaluate innate and adaptive responses are needed in order to broaden the possible clinical applications of immunological profiling in pneumonia. Improving the definitions of these new immunological phenotypes of CAP could contribute to the development of specific treatments.

In recent years, new insights into the pathogenic mechanisms of infectious lung disease have changed the previously accepted paradigm which held that the lungs were sterile. Multiple bacterial species are part of the healthy lung, forming what is known as the “lung microbiome” [83]. These groups of bacteria are part of a dynamic community, and their constant interaction with lung immunity maintains them in equilibrium in the healthy lung. This balance is disturbed by acute infections such as pneumonia [84] or chronic lung diseases such as non-cystic fibrosis bronchiectasis [85], COPD [86], or asthma [87]. This disequilibrium is known as “dysbiosis” and leads to significant modifications within the microbial communities in the healthy lung microbiome. To date, the risk factors for dysbiosis have not been defined, nor do we fully understand the ways in which bacteria cause disease and how imbalances occur [84].

Pneumococci are major bacterial pneumonia pathogens, and other species may be found in blood or pathologic lesions in some patients with severe CAP. The microbiota is composed of hundreds of types of commensal bacteria or normal flora. Mammals and their microbiota in the intestines, skin, lower urinary tract, upper respiratory tract, and possibly lower respiratory tract may have co-evolved. It is believed that these bacteria groups may initially have been external pathogens which then adapted to their hosts. Commensal bacteria may help to prevent the colonization of other external pathogens and provide the host with beneficial materials such as vitamins. In the mucosal immune system, gut-associated lymphoid tissues (GALT) are established after colonization of bacteria in the microbiota. These findings suggest that normal flora and host immune systems may communicate with each other. Normal flora may be less virulent than the initial external pathogens, though microbiota strains in the respiratory tract and in other regions can cause infections when they invade and replicate in the host. For example, pneumococcal bacteremia is not uncommon in febrile healthy young infants, but sepsis is very rare in immune-competent infants without antibiotic treatment.

The microbiota in an individual and in a species may change continuously due to environmental factors such as diet, antibiotic use, and adaptation of a pathogen to the host and population (herd immunity). It is known that the microbiota differs according to individual, ethnic group, and cultural environment. Antibiotic-resistant strains in the microbiota can spread to other populations via unknown mechanisms, even in people living in remote areas with no previous antibiotic use [88]. It is possible that respiratory strains that have potential for invading the host (cells) could differ across populations and change to milder strains over time. For example, before the early 20th century, scarlet fever caused by group A streptococci (GAS) was a severe disease with occasionally fatal outcomes in children; however, the severity and incidence of this disease and its complications, such as acute rheumatic fever and acute poststreptococcal glomerulonephritis, are now milder and less frequent. Today, the prevalence of GAS as well as *S. pneumoniae* in healthy children and adults is higher in these populations, suggesting that GAS and pneumococcus strains may have adapted to being normal flora of the respiratory tract [89,90]. Some respiratory pathogens that are resistant to antibiotics such as ampicillin, sulfa drugs, and quinolones have been well documented, as we note in this study [88]. However, these strains (including multidrug-resistant strains) may form part of the changing microbiota in individuals, and possibly in populations, during adaptation to the host; if so, it is possible that these strains may not act as external pathogens but as highly virulent internal pathogens in immune-competent hosts.

There are still major gaps in our knowledge of the role and functioning of the lung microbiome, such as the consequences of microbiome antimicrobial therapy, the alterations caused to the immune response, and its possible role in the prevention of diseases [91].

## 6. Raising Awareness about Pneumonia

Patient education is essential to reduce the incidence of pneumonia, especially in the major risk groups. It is important to share our knowledge of pneumonia, its causes, and methods for its prevention with the general public. Educational and awareness campaigns are urgently needed, especially now that we have vaccines that play a key role in the prevention of the disease. Last year, the Association of Support and Information for Family and Patients with Pneumonia (“NEUMOIA”, http://neumoai.org/) was created with the aim of spreading knowledge of this infectious disease and providing information that can help its prevention. This year, Barcelona will host the Global Forum on Childhood Pneumonia, which represents a major opportunity to ensure that pneumonia is at the forefront of national and global health agendas—especially considering that the disease kills one child under five years every 39 s [92]. Projects like this are likely to help to raise awareness about pneumonia among the general public and encourage them to keep their lungs healthy.

Although preventive vaccine strategies against influenza and pneumococci are important, many patients with CAP may be exposed to respiratory insults other than influenza. Future studies should be based on broadening our understanding of the immunopathogenesis of CAP, on the early and timely application of properly dose-adjusted immune modulator strategies based on the immunopathogenesis of pneumonia, and on strategies for early detection of pneumonia before disease progression, especially in elderly patients with limited immune function.

## 7. Conclusions

Pneumonia continues to be a major global health problem and a significant cause of morbidity and mortality worldwide. Despite the wealth of information on pneumonia currently available, some of the recommendations for the management of CAP in international clinical guidelines are still based on expert comments; in fact, no guideline updates have been issued in the past decade. The lack of specific evidence on the proper use of broad-spectrum antibiotics hampers the clinical management of pneumonia caused by MDR pathogens. Today, we have new technologies that could improve pneumonia management, but we must be sure to implement them in daily clinical practice.

The misperception of pneumonia among the general public is alarming. A great deal remains to be done to persuade people of the importance of healthy lifestyles and of the key role of vaccination for preventing pneumonia. To quote Chanderraj and Dickson [73] once again, pneumonia continues to be a 21st-century problem treated with 20th-century therapies and diagnosed using 19th-century tools.

This call is addressed to the health authorities, urging them to attach more importance to pneumonia and to promote international and national projects, and to the scientific societies whose task it is to guarantee high quality standards among health professionals. It is also aimed at practitioners who have contact with patients and their families, because the provider–patient relationship is fundamental to the attempts to raise the public’s awareness of the importance of CAP and to involve it in its prevention.

## Figures and Tables

**Figure 1 medsci-08-00006-f001:**
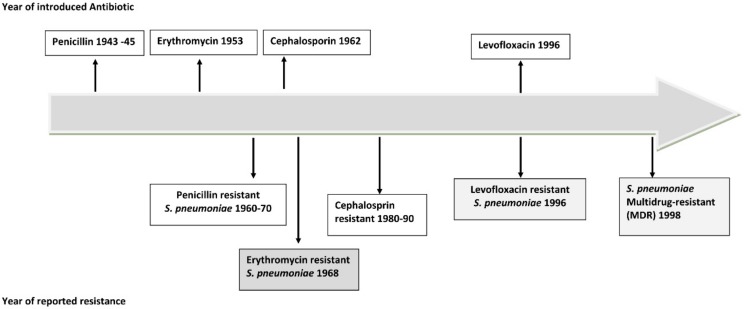
Timeline of antibiotic resistance of *Streptococcus pneumoniae* (pneumococcus).

**Figure 2 medsci-08-00006-f002:**
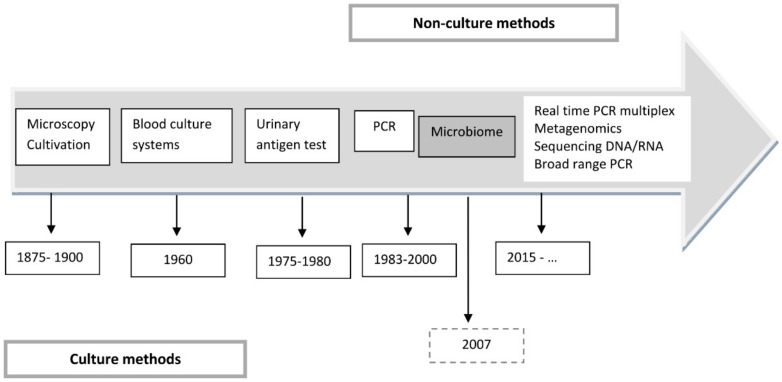
Evolution of diagnostic test for pneumonia.

**Table 1 medsci-08-00006-t001:** Recent studies of immunological profiles in community-acquired pneumonia.

Study	Immunological Profile	Clinical Correlate
Bermejo-Martin et al. [77] (*n* = 4396; CAP immunocompetent)	Lymphocytes (<724 cells/mm^3^) L-CAP phenotype	Increased ICU admission
		More complications
		Increased 30-day mortality
Mendez et al. [78] (*n* = 217, CAP immunocompetent)	L-CAP phenotype present: CD4^+^ depletion, higher inflammatory response, and low IgG2 levels	Increased severity of pneumonia at presentation
		Increased treatment failure
		Increased 30-day mortality
Güell et al. [79] (*n* = 710; CAP admitted to ICU)	Lymphopenia <675 cells/mm^3^ or <501 cells/mm^3^	2.32- and 3.76-fold risk of mortality in patients with or without septic shock
	Neutrophils <8850 cells/mm^3^	3.6-fold risk of mortality

Abbreviations: CAP: community-acquired pneumonia; ICU: intensive care unit; L-CAP: lymphopenic community-acquired pneumonia.

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
