# Peer review of "Defining Community-Acquired Pneumonia as a Public Health Threat: Arguments in Favor from Spanish Investigators"

_medsci, 2020, doi:10.3390/medsci8010006_

Round 1

Reviewer 1 Report

Overall - I expected this to be a thorough review of CAP that would go through etiology, global and local burden, progress (new vaccines, smoking cessation), failures (e.g. low vaccination rates) and next steps (I still don't know what those are from this paper). I did not get the feeling the flow of this paper was well enough thought out. Further, to make CAP a "public health threat" one needs some relative burden comparisons for which there are several references of how CAP compares to other serious disease (e.g. heart disease, stroke).

I may reject primarily due to the lack of thoroughness, grammar throughout (need English editing), and, despite 80+ references, several key ones missing.

Should it be a a call "from Spain" or "in Spain"?

Comments like "pneumonia is still" in abstract need to be justified. Perhaps, "Despite advances in prevention..." as a justification.

Lines 86-89 are redundant and out of place and can be moved up to earlier in introduction or deleted. 

All section headings need editing. Most don't make sense grammatically. Others just don't make sense, "When negative alliances cause devastation"

Author Response

Reviewer 1

Overall, I expected this to be a thorough review of CAP that would go through etiology, global and local burden, progress (new vaccines, smoking cessation), failures (e.g. low vaccination rates) and next steps (I still don't know what those are from this paper). I did not get the feeling the flow of this paper was well enough thought out. Further, to make CAP a "public health threat" one needs some relative burden comparisons for which there are several references of how CAP compares to other serious disease (e.g. heart disease, stroke). I may reject primarily due to the lack of thoroughness, grammar throughout (need English editing), and, despite 80+ references, several key ones missing.

Answer: Thanks for the comment. Our article is not a review article. Our aim is to provide a concise summary of the impact of pneumonia and why we should attach more importance to this infectious disease. We have checked the grammar throughout the article.

Should it be a call "from Spain" or "in Spain"?

Answer: A call “from Spain” is correct, because the article presents the  point of view of a group of Spanish researchers regarding pneumonia.

Comments like "pneumonia is still" in abstract need to be justified. Perhaps, "Despite advances in prevention..." as a justification.

Answer: We have modified this information in the abstract.

Lines 86-89 are redundant and out of place and can be moved up to earlier in introduction or deleted. 

Answer: We have deleted the redundant information, as the reviewer suggested.

All section headings need editing. Most don't make sense grammatically. Others just don't make sense, "When negative alliances cause devastation"

Answer: We have edited all section headings and modified them according to the reviewer’s suggestion.

Reviewer 2 Report

In this paper, the authors reviewed regarding recent epidemiological situation and public health problems in community-acquired pneumonia (CAP) in Spain.

This article is well-written and discussed on some unsolved therapeutic issues in CAP, mainly resulted from the increase in the number of elderly patients with multiple comorbidities, antibiotic-resistant pathogens and the difficulty in a rapid diagnosis.

This reviewer suggests some new sights of view on the issues.

Introduction

The authors could add the results of other studies that initial insults of CAP may be more frequently caused by antibiotic non-response pathogens such as influenza viruses, SARS viruses, other respiratory viruses or mycoplasmas than bacterial pathogens. Severe pneumonia caused by the insult from these pathogens can induce secondary bacterial invasion such as S. pneumoniae, S. aureus and P. aeruginosa that may be colonized in the host respiratory tract. It is possible that bacterial detection in these patients, including patients with sepsis, may mean severe CAP from initial insults, especially in immune-incompetent persons or elderly persons with comorbidities. Thus, early antibiotic treatment may have a limited effect in the patients with severe CAP, since initial insults already established at presentation time may be responsible for extensive lung injury (please see below section). Furthermore, some patients with severe pneumonia and subsequent ARDS are caused by non-pathogen events such as blunt chest trauma, amniotic fluid embolism and aspiration, which can also induce a secondary bacterial invasion.

CAP affects everyone: why don't we recognize the threat?

This part is well-written, and it is impressive that severe CAP patients may have long-term and short-term undesirable consequences such as higher risk of cardiac events. This finding could be in part explained that the substances that are produced, including DAMPs, during severe lung tissue cell injury may affect on other host cells such as aggravation of heart cell injury.

When negative alliances cause devastation

As for 2009 H1N1 pandemic influenza, it is true that old children group and young adult groups (20-40s) were mainly affected with a rapidly progressive severe pneumonia, opposite to seasonal influenza (mortality is higher in infants and elderly persons). Now, it has been recognized that antiviral agents for influenza infection, such as oseltamivir, have a limited effect on prevention of pneumonia or progression of pneumonia (Jefferson T, et al. BMJ. 2014;348:g2545). Through the studies on 2009 H1N1 pandemic epidemiological and clinical characteristics, we have learned that the pathogenesis of influenza is not a virus-induced cytopathy but a hyperimmune reaction of the host (such as cytokine storm) against the insults from influenza virus infection. In addition, some investigators have suggested that timely early-immune modulator therapy was effective to prevent from the disease progression in the 2009 pandemic influenza.

For the issues of polymicrobial CAP and the relationship between influenza and S. pneumoniae and other pathogens, please read the reviewer’s above described comments for the introduction.

With technology awaiting us, a step forward is needed

The authors described that limitations of PCR method such as the difficulty in discriminating between colonization and infection in many microorganisms other than pneumococcus. However, pneumococcus now is frequently colonized in healthy children and adults in many countries as possibly becoming a normal flora of upper respiratory tract (Wang L, et al. BMC Infect Dis. 2017;17(1):765).

They could add the limitations of diagnostic methods for pneumonia in the present time; detection of agents on upper or lower respiratory tract is not a direct etiologic substances of lung cell injury in pneumonia and does not reflect a systemic immune reaction against the pathogens, since few whole agents have been discovered in pathologic lesions of lungs except secondary invasive bacteria in fatal cases in viral pneumonia (influenza).

New concepts, when times change

This reviewer agrees with the concept that patients with lymphopenic CAP may have severe pneumonia and poor prognosis. The authors could provide more explanations or their opinions regarding the reason of appearance of lymphopenia, immunopathogenesis of CAP, and the rationale of early immune-modulator treatment for the prospective of this study.

Following comments may be helpful for revision of the paper.

- Lymphopenia or eventual leukopenia may be characteristic of severe pneumonia patients infected with respiratory pathogens, including influenza viruses, corona viruses, measles virus, and Mycoplasma pneumoniae. The severity of lymphopenia is correlated with the severity of lung injury. The autopsy findings of severe ARDS patients and experimental animals infected with influenza viruses show lymphocyte depletion of whole lymphoid tissues. This finding, together with lymphocyte predominance in early lung lesions, suggests that T cells may control the substances from pathogens and/or injured host cells. Animals with depressed T cell function or loss of T cell function such as nude mice show milder or few pneumonia lesions in comparison to immune-competent animals in mycoplasma or influenza virus infection models, although the duration of pathogen detection in the lungs of animals with compromised T cells is longer. These finding suggest that lung injury is associated with T cell activation rather than with pathogen inducing cytopathy. It is possible that there is a limitation on the numerical capacity of the host immune system on mobilizing immune cells against these relentless substances to counter extensive lung cell injury in immune-competent patients. Patients with underlying diseases or immune-deficient states may have a limited repertoire of immune cells. Furthermore, severe pneumonia or ARDS from a viral infection tends to induce subsequent bacterial infection from normal flora in patients, which adds to the workload of immune cells (Lee KY. Int J Mol Sci 2017;18(2): E388).

- Although immunopathogenesis of pneumonia, as one of infectious diseases or other conditions, remains unknown, it is proposed that there are inflammation-inducing substances in each pneumonia. Host immune system may control these etiologic substances not only those originate from pathogens, including toxins and pathogen-associated molecular patterns (PAMPs), but also those originated from injured infected-host cells including damage-associated molecular patterns (DAMPs), pathogenic proteins, and pathogenic peptides, especially in intracellular pathogen infections such as virus, chlamydia, legionella and possibly mycoplasma infections (the protein-homeostasis-system hypothesis). When these substances spread systemically and locally and bind to target organ cells, clinical symptoms begin due to the activation of corresponding immune cells and immune proteins. The substances produced from injured host cells induce further inflammation if released into the systemic circulation or near local lesions. The severity or chronicity of pneumonia or ARDS depends on the amount of etiologic substances with corresponding immune reactions, the duration of the appearance of specific immune cells, or the repertoire of specific immune cells that control the substances. Therefore, early control of lung injuries from initial hyperactive immune reactions performed by non-specific adaptive immune cells is crucial for reduction of morbidity and prevention of pneumonia progression in patients with CAP including severe influenza pneumonia, mycoplasma pneumonia, and other types of pneumonia. Thus, it is proposed that earlier dose-adjusted corticosteroid therapy with an antibiotic can reduce morbidity of pneumonia and prevent disease progression (Yang EA, et al. J Clin Med 2019;8:726).

- About dysbiosis;

At the present time, primary bacterial CAP is rare in the developed countries, and its contagiousness to other persons is very low because majority of possible bacterial agents of CAP may be strains in microbiota of the same species (human).

Pneumococci are major bacterial pneumonia pathogen, and other species can be found in blood or pathologic lesions in some patients with severe CAP. Microbiota is composed of hundreds of types of commensal bacteria or normal flora. Mammals and their microbiota in the intestines, skin, lower urinary tract, upper respiratory tract and possibly lower respiratory tract may have co-evolved. It is believed that these bacteria groups may have initially been external pathogens, and then they have adapted with hosts. Commensal bacteria may help to prevent from colonization of other external pathogens and provide some beneficial materials such as vitamins for the host. In the mucosal immune system, gut-associated lymphoid tissues (GALT) were established after colonization of bacteria in microbiota. These findings suggest that normal flora and host immune systems may communicate with each other in some aspect. Thus, normal flora may be less virulent compared to the initial external pathogens, though microbiota strains in respiratory tract and in other regions can cause infections when they invade and replicate in the host. For example, pneumococcal bacteremia is not uncommon in febrile healthy young infants, but sepsis is very rare in immune-competent infants without antibiotic treatment.

The microbiota in an individual and in a species may continuously change due to environmental factors such as diets, antibiotic use, and adaptation of a pathogen to the host and population (herd immunity). It is known that microbiota differs in different individuals, ethnic groups, and cultural environments. Antibiotic-resistant strains in microbiota can spread to other populations via unknown mechanisms, including people living in a remote area with no previous antibiotic use (Pallecchi L, et al. Expert Rev Anti Infect Ther 2008;6:725-32). It is possible that respiratory strains that have a potential to invade into the host (cells) could differ across the populations and change to milder strains over time. For example, before the early 20th century, scarlet fever caused by group A streptococci (GAS) was a severe disease with occasional fatal cases in children. However, the severity and incidence of scarlet fever and its complications such as acute rheumatic fever and acute poststreptococcal glomerulonephritis are now milder and lower. Now, the prevalence of GAS carriers as well as S. pneumoniae in healthy children and adults is higher in the populations, suggesting that GAS strains and pneumococcus strains in respiratory tract may have adapted to being normal flora of the respiratory tract. Some respiratory pathogens that are resistant to antibiotics such as ampicillin, sulfa drugs and quinolones have been well documented as the authors reviewed in this study. But, these strains, including multi-drug resistant strains, may be the changing microbiota in individuals and possibly in the populations during adaptation to hosts. Therefore, it is possible that these strains may not act as external pathogens with high virulence in immune-competent hosts.

When the patient's opinion matters

Although preventive vaccine strategies against influenza and pneumococci is important, many patients with CAP may have other respiratory insults other than influenza. The understanding of immunopathogenesis of CAP, and early and timely application of properly dose-adjusted immune modulator strategies based on the immunopathogenesis of pneumonia, and strategies for early detection of pneumonia before disease progression, especially in elderly patients having a limited immune function may remain in future studies.

Author Response

Reviewer 2

In this paper, the authors reviewed regarding recent epidemiological situation and public health problems in community-acquired pneumonia (CAP) in Spain. This article is well-written and discussed on some unsolved therapeutic issues in CAP, mainly resulted from the increase in the number of elderly patients with multiple comorbidities, antibiotic-resistant pathogens and the difficulty in a rapid diagnosis. This reviewer suggests some new sights of view on the issues.

Answer: We have modified the article in accordance with the reviewer’s suggestions.

Introduction

The authors could add the results of other studies that initial insults of CAP may be more frequently caused by antibiotic non-response pathogens such as influenza viruses, SARS viruses, other respiratory viruses or mycoplasmas than bacterial pathogens. Severe pneumonia caused by the insult from these pathogens can induce secondary bacterial invasion such as S. pneumoniae, S. aureus and P. aeruginosa that may be colonized in the host respiratory tract. It is possible that bacterial detection in these patients, including patients with sepsis, may mean severe CAP from initial insults, especially in immune-incompetent persons or elderly persons with comorbidities. Thus, early antibiotic treatment may have a limited effect in the patients with severe CAP, since initial insults already established at presentation time may be responsible for extensive lung injury (please see below section). Furthermore, some patients with severe pneumonia and subsequent ARDS are caused by non-pathogen events such as blunt chest trauma, amniotic fluid embolism and aspiration, which can also induce a secondary bacterial invasion.

Answer: We have added some of the results of the studies that the reviewer suggests. Page 1, line 43-48 .

CAP affects everyone: why don't we recognize the threat?

This part is well-written, and it is impressive that severe CAP patients may have long-term and short-term undesirable consequences such as higher risk of cardiac events. This finding could be in part explained that the substances that are produced, including DAMPs, during severe lung tissue cell injury may affect on other host cells such as aggravation of heart cell injury.

 Answer: We have added the comment that the reviewer suggests. Page 3, line 129-132.

When negative alliances cause devastation

As for 2009 H1N1 pandemic influenza, it is true that old children group and young adult groups (20-40s) were mainly affected with a rapidly progressive severe pneumonia, opposite to seasonal influenza (mortality is higher in infants and elderly persons). Now, it has been recognized that antiviral agents for influenza infection, such as oseltamivir, have a limited effect on prevention of pneumonia or progression of pneumonia (Jefferson T, et al. BMJ. 2014;348:g2545). Through the studies on 2009 H1N1 pandemic epidemiological and clinical characteristics, we have learned that the pathogenesis of influenza is not a virus-induced cytopathy but a hyperimmune reaction of the host (such as cytokine storm) against the insults from influenza virus infection. In addition, some investigators have suggested that timely early-immune modulator therapy was effective to prevent from the disease progression in the 2009 pandemic influenza. For the issues of polymicrobial CAP and the relationship between influenza and S. pneumoniae and other pathogens, please read the reviewer’s above described comments for the introduction.

Answer: We have added the comment that the reviewer suggests. Page 4, line 171-174 and line 180 – 183.

With technology awaiting us, a step forward is needed

The authors described that limitations of PCR method such as the difficulty in discriminating between colonization and infection in many microorganisms other than pneumococcus. However, pneumococcus now is frequently colonized in healthy children and adults in many countries as possibly becoming a normal flora of upper respiratory tract (Wang L, et al. BMC Infect Dis. 2017;17(1):765).They could add the limitations of diagnostic methods for pneumonia in the present time; detection of agents on upper or lower respiratory tract is not a direct etiologic substances of lung cell injury in pneumonia and does not reflect a systemic immune reaction against the pathogens, since few whole agents have been discovered in pathologic lesions of lungs except secondary invasive bacteria in fatal cases in viral pneumonia (influenza).

Answer: We have modified the section according to the reviewer’s suggestion. Page 5, line 127-129

New concepts, when times change

This reviewer agrees with the concept that patients with lymphopenic CAP may have severe pneumonia and poor prognosis. The authors could provide more explanations or their opinions regarding the reason of appearance of lymphopenia, immunopathogenesis of CAP, and the rationale of early immune-modulator treatment for the prospective of this study. Following comments may be helpful for revision of the paper.

- Lymphopenia or eventual leukopenia may be characteristic of severe pneumonia patients infected with respiratory pathogens, including influenza viruses, corona viruses, measles virus, and Mycoplasma pneumoniae. The severity of lymphopenia is correlated with the severity of lung injury. The autopsy findings of severe ARDS patients and experimental animals infected with influenza viruses show lymphocyte depletion of whole lymphoid tissues. This finding, together with lymphocyte predominance in early lung lesions, suggests that T cells may control the substances from pathogens and/or injured host cells. Animals with depressed T cell function or loss of T cell function such as nude mice show milder or few pneumonia lesions in comparison to immune-competent animals in mycoplasma or influenza virus infection models, although the duration of pathogen detection in the lungs of animals with compromised T cells is longer. These finding suggest that lung injury is associated with T cell activation rather than with pathogen inducing cytopathy. It is possible that there is a limitation on the numerical capacity of the host immune system on mobilizing immune cells against these relentless substances to counter extensive lung cell injury in immune-competent patients. Patients with underlying diseases or immune-deficient states may have a limited repertoire of immune cells. Furthermore, severe pneumonia or ARDS from a viral infection tends to induce subsequent bacterial infection from normal flora in patients, which adds to the workload of immune cells (Lee KY. Int J Mol Sci 2017;18(2): E388).

- Although immunopathogenesis of pneumonia, as one of infectious diseases or other conditions, remains unknown, it is proposed that there are inflammation-inducing substances in each pneumonia. Host immune system may control these etiologic substances not only those originate from pathogens, including toxins and pathogen-associated molecular patterns (PAMPs), but also those originated from injured infected-host cells including damage-associated molecular patterns (DAMPs), pathogenic proteins, and pathogenic peptides, especially in intracellular pathogen infections such as virus, chlamydia, legionella and possibly mycoplasma infections (the protein-homeostasis-system hypothesis). When these substances spread systemically and locally and bind to target organ cells, clinical symptoms begin due to the activation of corresponding immune cells and immune proteins. The substances produced from injured host cells induce further inflammation if released into the systemic circulation or near local lesions. The severity or chronicity of pneumonia or ARDS depends on the amount of etiologic substances with corresponding immune reactions, the duration of the appearance of specific immune cells, or the repertoire of specific immune cells that control the substances. Therefore, early control of lung injuries from initial hyperactive immune reactions performed by non-specific adaptive immune cells is crucial for reduction of morbidity and prevention of pneumonia progression in patients with CAP including severe influenza pneumonia, mycoplasma pneumonia, and other types of pneumonia. Thus, it is proposed that earlier dose-adjusted corticosteroid therapy with an antibiotic can reduce morbidity of pneumonia and prevent disease progression (Yang EA, et al. J Clin Med 2019;8:726).

- About dysbiosis;

At the present time, primary bacterial CAP is rare in the developed countries, and its contagiousness to other persons is very low because majority of possible bacterial agents of CAP may be strains in microbiota of the same species (human).

Pneumococci are major bacterial pneumonia pathogen, and other species can be found in blood or pathologic lesions in some patients with severe CAP. Microbiota is composed of hundreds of types of commensal bacteria or normal flora. Mammals and their microbiota in the intestines, skin, lower urinary tract, upper respiratory tract and possibly lower respiratory tract may have co-evolved. It is believed that these bacteria groups may have initially been external pathogens, and then they have adapted with hosts. Commensal bacteria may help to prevent from colonization of other external pathogens and provide some beneficial materials such as vitamins for the host. In the mucosal immune system, gut-associated lymphoid tissues (GALT) were established after colonization of bacteria in microbiota. These findings suggest that normal flora and host immune systems may communicate with each other in some aspect. Thus, normal flora may be less virulent compared to the initial external pathogens, though microbiota strains in respiratory tract and in other regions can cause infections when they invade and replicate in the host. For example, pneumococcal bacteremia is not uncommon in febrile healthy young infants, but sepsis is very rare in immune-competent infants without antibiotic treatment.

The microbiota in an individual and in a species may continuously change due to environmental factors such as diets, antibiotic use, and adaptation of a pathogen to the host and population (herd immunity). It is known that microbiota differs in different individuals, ethnic groups, and cultural environments. Antibiotic-resistant strains in microbiota can spread to other populations via unknown mechanisms, including people living in a remote area with no previous antibiotic use (Pallecchi L, et al. Expert Rev Anti Infect Ther 2008;6:725-32). It is possible that respiratory strains that have a potential to invade into the host (cells) could differ across the populations and change to milder strains over time. For example, before the early 20th century, scarlet fever caused by group A streptococci (GAS) was a severe disease with occasional fatal cases in children. However, the severity and incidence of scarlet fever and its complications such as acute rheumatic fever and acute poststreptococcal glomerulonephritis are now milder and lower. Now, the prevalence of GAS carriers as well as S. pneumoniae in healthy children and adults is higher in the populations, suggesting that GAS strains and pneumococcus strains in respiratory tract may have adapted to being normal flora of the respiratory tract. Some respiratory pathogens that are resistant to antibiotics such as ampicillin, sulfa drugs and quinolones have been well documented as the authors reviewed in this study. But, these strains, including multi-drug resistant strains, may be the changing microbiota in individuals and possibly in the populations during adaptation to hosts. Therefore, it is possible that these strains may not act as external pathogens with high virulence in immune-competent hosts.

Answer: We thank the reviewer for these helpful suggestions, which we have incorporated in the corresponding sections. Page 6, line 247-282, and Page 7, line 297 – 328.

When the patient's opinion matters

Although preventive vaccine strategies against influenza and pneumococci is important, many patients with CAP may have other respiratory insults other than influenza. The understanding of immunopathogenesis of CAP, and early and timely application of properly dose-adjusted immune modulator strategies based on the immunopathogenesis of pneumonia, and strategies for early detection of pneumonia before disease progression, especially in elderly patients having a limited immune function may remain in future studies.

Answer: We have modified the section according to the reviewer’s suggestions. Page 8, line 342-347.

Round 2

Reviewer 1 Report

I don’t believe this to be of sufficient quality, depth, or perspective for publication. 

Author Response

Answer to Reviewer 1:

Many thanks for the comments of the reviewer. We believe that this article is very important, especially considering the incidence and mortality of pneumonia, a infectious disease that can be prevented. The article has been completely reviewed by an English expert, as suggested by the reviewer,